# WAVE: Window-Aware Vocabulary-Efficient Early-Exit for Training-Free LLM Acceleration

Seonggeun Kim [1]   Gilha Lee [1]   Hyun Kim [1]

## Abstract

Large language models (LLMs) incur substantial inference latency due to autoregressive decoding, in which each token requires a full forward pass through all transformer layers. Early-exit methods that terminate computation at intermediate layers offer a promising remedy, yet existing approaches suffer from fundamental limitations. Confidence-based methods rely on evaluating the full LM head at every layer, introducing considerable overhead that can negate the expected speedup. Schedule-based methods avoid this cost through predetermined exit schedules, but their monotonically decreasing layer allocation collapses to shallow layers, thereby constraining the maximum generation length. Learned exit predictors further require costly task-specific training and are vulnerable to distribution shifts in unseen domains. We propose Window-Aware Vocabulary-Efficient Early-Exit (**WAVE**), a training-free framework that addresses these challenges through two key innovations. First, *exit window scheduling* identifies an optimal layer range for early-exit decisions via offline calibration, preventing premature convergence to shallow layers while substantially reducing the number of exit checks. Second, a *proxy LM head* constructs a lightweight vocabulary subset at the window's starting layer, reducing per-layer exit overhead by **87%** relative to full LM head. WAVE requires no gradient-based training and enables immediate deployment with only a brief calibration phase. Experiments on LLaMA-2 7B demonstrate up to **1.4×** average speedup while preserving output quality, with full compatibility with W4A16 quantization, establishing WAVE as

a practical early-exit framework for accelerating LLMs inference without retraining.

## 1. Introduction

Large language models (LLMs) have demonstrated remarkable performance in natural language understanding and generation (Grattafiori et al., 2024; Achiam et al., 2023; Bai et al., 2023). However, achieving such performance requires billions of parameters and incurs substantial computational costs (Wan et al., 2024). In particular, the LLM inference process, which operates through autoregressive decoding, repeatedly performs a full model forward pass to generate each token. During this process, all input tokens undergo the same computation depth (*i.e.*, the number of transformer layers actually executed per token), regardless of their difficulty or information content. As shown in Figure 1, this sequential process structure and fixed computation depth are primary causes of redundant computation during inference, significantly increasing inference latency. This inference bottleneck directly results in reduced throughput and increased service costs in real-world deployment, severely hindering scalability and cost-effectiveness in long-context settings. Consequently, there is increasing demand for inference efficiency techniques that adapt computation to input characteristics, addressing the limitations of static inference methods that apply uniform computation across all inputs.

LLM inference efficiency techniques can be broadly classified into two paradigms. The first is the model compression approach, which reduces computational and memory footprints by compressing the model size through quantization, pruning, and knowledge distillation (Kang & Kim, 2025; Kang et al., 2025; Choi & Kim, 2025; Kim & Kim, 2023; Koo & Kim, 2023; Bercovich et al., 2025; Zhang et al., 2024; Dasgupta & Cohn, 2025; Hong & Kim, 2025). While these techniques provide consistent acceleration without runtime overhead, they still have the limitation of applying the same computation depth regardless of input difficulty. The second paradigm is adaptive decoding, which dynamically adjusts the computation at inference time based on input characteristics. Adaptive decoding strategies can be categorized into those employing auxiliary models to decode multiple

---

[1]Department of Electrical and Information Engineering and Research Center for Electrical and Information Technology, Seoul National University of Science and Technology, Seoul, Republic of Korea. Correspondence to: Hyun Kim <hyunkim@seoultech.ac.kr>.

*Proceedings of the 43$^{rd}$ International Conference on Machine Learning*, Seoul, South Korea. PMLR 306, 2026. Copyright 2026 by the author(s).

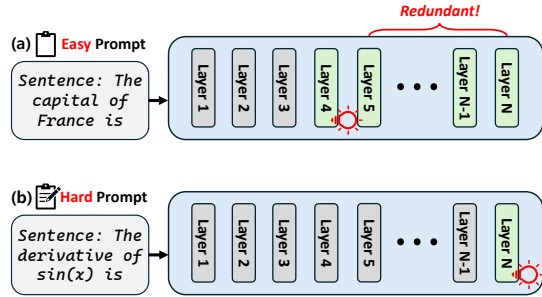

*Figure 1.* Inefficiency of autoregressive decoding under a uniform computation depth. (a) Easy predictions reach high confidence early, making later layers redundant. (b) Hard predictions require deeper layers to form sufficient confidence.

tokens simultaneously and those adjusting per-layer computation depth within a single model. Speculative decoding (SD) (Chen et al., 2023) falls into the former category; it utilizes a lightweight draft model to autoregressively generate candidate tokens, which are then verified in parallel by the target model. While SD offers high acceleration potential by validating multiple tokens in a single step and preserving the target model's output distribution, it incurs additional training costs for the draft model and memory overhead due to separate key-value (KV) caches. Consequently, early-exit (EE) methods have emerged as a promising alternative. EE adaptively adjusts computation depth within a single LLM by skipping subsequent layers when intermediate representations are deemed sufficient under an exit policy. This approach incurs no additional memory overhead, as it operates solely on the original model. Even when an exit predictor is employed, its training cost is substantially lower than that of draft models used in speculative decoding. Moreover, EE integrates seamlessly into existing serving pipelines and remains orthogonal to model compression techniques, enabling complementary use without interference.

Despite these advantages, existing EE methods suffer from approach-specific limitations. Confidence-based methods (Schuster et al., 2022; Bae et al., 2023) rely on frequent logit computation via the language modeling (LM) head and use probability-based metrics (*e.g.*, softmax, entropy, margin) at intermediate layers, which can substantially offset the computational savings of early exiting and, in some cases, even increase inference latency. Schedule-based methods (Del Corro et al., 2023; Fan et al., 2025a) avoid this overhead by fixing the maximum layer usage according to predefined, position-dependent decay rules. However, monotonically decreasing schedules often converge prematurely to shallow layers, limiting generation length and degrading output quality. Predictor-based methods (Xu et al., 2025; Fan et al., 2025b) employ lightweight MLPs to infer exit decisions from intermediate features, but incur additional data collection and training costs and require retraining under domain shifts.

In this work, we propose WAVE, a Window-Aware Vocabulary-Efficient Early-Exit framework, to address the above limitations. The main contributions are as follows:

- **Mitigating Exit Overhead via Vocabulary-Efficient Proxy LM Head**: To reduce the excessive overhead caused by per-layer full LM head calls in confidence-based EE, we perform the full LM head only once at the starting layer of the exit window to extract the top-$K$ vocabulary, and then estimate confidence using only the proxy LM head thereafter. Through this, WAVE enables stable exit decisions while significantly reducing exit decision costs.

- **Preventing Shallow-Schedule Convergence with Calibrated Window-Aware Exit**: To alleviate the problem of schedule-based EE's monotonically decreasing schedules converging to shallow layers early on and limiting the number of tokens that can be generated, we define the layer interval where confidence transitions are concentrated as a window through offline calibration, and allow exits only within that window during inference. Through this, WAVE maintains stable generation quality while guaranteeing a layer range where meaningful representations are formed.

- **Training-free and Robust Deployment without Draft Models or Exit Predictors**: WAVE can be immediately deployed by constructing the exit schedule and proxy head using only calibration data from the original model, without a draft model or exit predictor training. Additionally, it can be applied without additional data collection or retraining when the domain changes, enhancing practicality.

As a result, the proposed method achieves an average inference speedup of approximately $1.4\times$ while maintaining performance degradation below 1%p across a wide range of language understanding and reasoning benchmarks. In particular, WAVE can be applied using only a single model without any auxiliary components, making it a practical acceleration strategy for reducing both latency and resource consumption in real-world operational settings.

## 2. Background

### 2.1. Dynamics of Layer-wise Representations in LLMs

In the autoregressive generation of LLMs, each token sequentially passes through $L$ transformer layers. Let $\mathbf{h}_l \in \mathbb{R}^d$ denote the hidden state of layer $l$, where $d$ is the hidden dimension and $\mathbf{h}_0$ is the input embedding or the output from the previous layer. Each layer is expressed as follows:

$$\mathbf{h}_l = \text{Layer}_l(\mathbf{h}_{l-1}), \quad l = 1, \dots, L \quad (1)$$

The output $\mathbf{h}_L$ of the final layer $L$ is projected through the LM head into a logit vector of size $V$, where $V$ is the vocabulary size of the LLM. Let $W_{\text{lm}} \in \mathbb{R}^{V \times d}$ denote the weight matrix of the LM head. The logit $\mathbf{z}$ and the probability distribution $p(y)$ of the next token are computed as:

$$\mathbf{z} = W_{\text{lm}}\mathbf{h}_L, \quad p(y) = \text{softmax}(\mathbf{z}) \tag{2}$$

This autoregressive formulation implicitly assumes that all $L$ transformer layers contribute equally to next-token prediction. However, recent analyses such as Logit Lens (nostalgebraist, 2020) reveal that transformer layers exhibit non-uniform roles: lower and middle layers primarily form representations, while upper layers refine predictions. This layer-wise specialization suggests that the depth at which sufficient information is available varies across tokens, indicating that applying a fixed computation depth to all inputs is computationally inefficient (Schuster et al., 2022; Elhoushi et al., 2024).

Building on this theoretical insight, our study investigates the validity of EE by examining the relationship between internal representation dynamics and prediction confidence in practical LLMs. Recent studies (Skean et al., 2025) show that decoder-only LLMs do not accumulate representations uniformly across layers but instead undergo distinct representation transitions within specific middle-layer intervals. This behavior is also reflected in prediction confidence trends, as illustrated in Figure 2. On multiple-choice benchmarks such as MMLU and CSQA, the probability of the correct token increases sharply upon entering the middle-layer interval, while the confidence gap over incorrect candidates widens substantially. In contrast, later layers largely preserve this high confidence with minimal changes in relative ranking. These observations indicate that the critical decision for next-token prediction is effectively resolved within a confined confidence formation interval, well before the final layer. Consequently, executing EE at intermediate layers ($l < L$) where confidence has sufficiently stabilized allows redundant computation to be skipped without degrading output quality. For instance, in LLaMA-2 7B with 32 layers, exiting after layer 22 yields approximately a 31% reduction in computation. This highlights the importance of precisely identifying and targeting the layer interval where confidence transitions are concentrated to maximize EE effectiveness.

## 2.2. Speculative Decoding and its Resource Overheads

SD reduces the number of target model invocations by letting a lightweight draft model generate multiple candidate tokens in advance, which are then verified in parallel by a single forward pass of the target model via rejection sampling. Given $k$ draft tokens $\hat{y}_1, \ldots, \hat{y}_k$ produced by the draft model $M_d$, the target model $M_t$ evaluates their probabilities and accepts valid tokens jointly. Medusa (Cai et al., 2024)

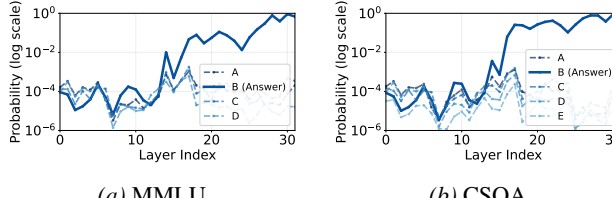

*(a)* MMLU  *(b)* CSQA

*Figure 2.* Layer-wise confidence dynamics of answer candidates under **LLaMA-2 7B** on two multiple-choice tasks: (a) MMLU and (b) CommonsenseQA (CSQA).

eliminates the need for an external draft model by attaching multiple auxiliary heads to the target model to predict future tokens in parallel. EAGLE (Li et al., 2024) trains an auto-regressive draft model using second-to-last-layer features of the target model, exploiting the observation that feature-level prediction is easier than direct token prediction. DistillSpec (Zhou et al., 2023) further aligns the draft and target models through knowledge distillation with on-policy data and task-specific divergence objectives.

While SD methods offer theoretically sound acceleration without degrading output quality, they incur substantial overhead from training auxiliary heads or draft models and maintaining additional parameters and KV caches. Moreover, retraining and realignment requirements limit deployment flexibility, particularly in dynamic or resource-constrained serving environments.

## 2.3. Existing Early-Exit Strategies and Limitations

While SD requires auxiliary draft components and incurs additional training and memory overhead, EE accelerates inference by dynamically reducing the average computation depth within a single model based on per-token difficulty. As discussed in Section 2.1, some tokens can be predicted reliably using representations from shallow layers, whereas others require deeper computation for complex reasoning.

Existing EE methods can be broadly categorized into three classes based on their exit decision mechanisms. Confidence-based methods estimate prediction confidence at intermediate layers and exit when the output distribution is sufficiently decisive. CALM (Schuster et al., 2022) employs softmax-based confidence metrics, while FREE (Bae et al., 2023) adaptively determines exit thresholds. However, repeated confidence evaluation across layers introduces non-trivial overhead, which can offset the computational gains of EE. Schedule-based methods avoid confidence computation by predefining the computation depth as a function of decoding progress. SkipDecode (Del Corro et al., 2023) applies a linear decay based on token position, and $D^3$ (Fan et al., 2025a) proposes a position-aware power-law decay. While efficient, monotonically decreasing schedules may force early exits before sufficient representations are formed, de-

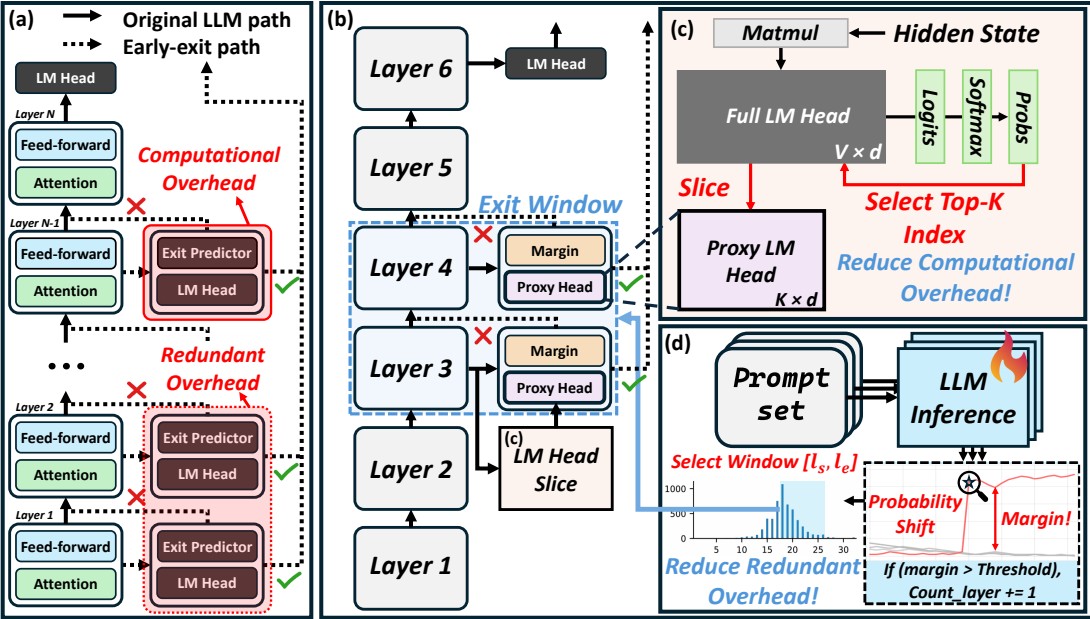

Figure 3. Comparison between prior EE approaches and WAVE. Frequent layer-wise exit checks cause redundant LM-head overhead, whereas WAVE restricts exiting to an offline-selected *exit window* and uses a top-$K$ proxy LM head for margin-based decisions.

grading generation quality or limiting the achievable generation length. Exit predictor-based methods train lightweight predictors using per-layer representations or logit-based features to determine exit decisions. AdaInfer (Fan et al., 2025b) and SpecEE (Xu et al., 2025) adopt classifier- or MLP-based predictors to enable adaptive exits. Despite their flexibility, they require additional data collection and training, are sensitive to predictor errors, and often need re-tuning when the domain shifts. In summary, existing EE methods suffer from at least one of the following limitations: overhead from confidence estimation, restrictive generation schedules, or additional training and maintenance costs.

## 3. Proposed Method

### 3.1. WAVE Framework Overview

Figure 3(a) shows that unnecessary overhead can accumulate due to repeated evaluation of exit conditions in existing EE methods. To address this, this paper proposes the WAVE framework as shown in Figure 3(b). The core idea of WAVE is to restrict the layer interval for exit decisions to an exit window and to reduce the cost of exit decisions within that interval. First, as shown in Figure 3(d), in the offline calibration stage, we observe per-layer confidence on a small set of calibration prompts and select an exit window $\mathcal{W}^{\star} = [l_s, l_e]$, where $l_s$ and $l_e$ denote the start and end layers of the window, respectively. Then, in online inference, exit decisions are made only within the configured exit window, reducing unnecessary repeated evaluations. Additionally, as shown in Figure 3(c), WAVE uses the top-$K$ vocabulary from the

initial exit layer to reduce computation in subsequent layers. For example, in Figure 3(b), exit decisions are performed only in the consecutive interval $[3, 4]$ of the middle layers. Furthermore, only the rows of the full LM head corresponding to the top-$K$ vocabulary obtained at the exit window starting layer 3 are sliced to construct a proxy LM head, which is then used for confidence computation.

### 3.2. Calibrating Window-Aware Exit Schedules

Performing exit decisions at every layer leads to excessive overhead due to repeated LM head-based confidence computation at each layer, while monotonically decreasing the exit layer with a fixed schedule causes convergence to early shallow layers, resulting in constraints on meaningful token generation. In this paper, we alleviate this trade-off by determining the window for exit decisions in advance through offline calibration. Specifically, after performing inference on calibration data, for each token $t$, we record the layer $l$ where the softmax margin $\Delta_l^{(t)} = p_l^{(1)} - p_l^{(2)}$ first exceeds the threshold $\tau_m$ as the hit layer $h_t$:

$$h_t = \min\{l : \Delta_l^{(t)} \geq \tau_m\} \tag{3}$$

Figure 4 visualizes the calibration results collected while generating 5K tokens using the GSM8K (Cobbe et al., 2021) dataset, aggregating the hit layer distribution for each token. The results show that hit layers tend to concentrate in a specific interval, with hit frequency dropping sharply outside that interval. This indicates that the layers requiring exit decisions are concentrated in a limited range, suggesting

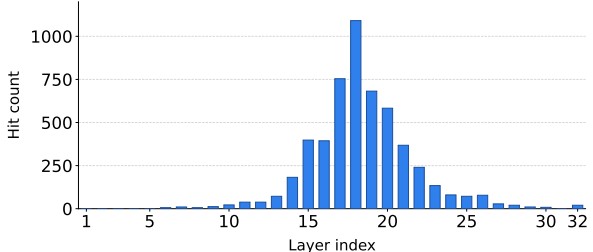

*Figure 4.* Offline calibration on GSM8K (5k tokens): distribution of hit layers $h_t = \min\{l : \Delta_l^{(t)} \geq \tau_m\}$ with $\tau_m = 0.3$.

that performing exit decisions at every layer is inefficient. Based on this observation, we restrict the layers for exit decisions to an exit window of fixed length $W$. This controls overhead by limiting the number of exit decisions to at most $W$, while preventing accuracy degradation by setting the window to sufficiently cover the hit distribution. Since exiting at too early layers can cause quality degradation due to error propagation, we set the center of the hit distribution as the starting point $l_s$. We define the window as $\mathcal{W}^\star = [l_s, l_e]$ to evaluate exit at most $W$ times per token, where $l_e = l_s + W$, and $T_{\text{cal}}$ denotes the total number of tokens observed during calibration:

$$l_s = \left\lfloor \frac{1}{T_{\text{cal}}} \sum_{t=1}^{T_{\text{cal}}} h_t \right\rfloor, \qquad \mathcal{W}^\star = [l_s, l_e] \qquad (4)$$

Subsequently, in online inference, exit decisions are performed only within the exit window $\mathcal{W}^\star$ determined offline. If the margin condition is satisfied within the window, exit occurs immediately; otherwise, the model continues through subsequent layers and generates output at the final layer $L$. Through this, the proposed method limits the number of exit decisions to at most $W$ compared to naive EE while ensuring stable token generation.

### 3.3. Vocabulary-Efficient Inference via Proxy LM Head

Existing confidence-based methods compute token-level confidence at every layer for exit decisions. This involves the full LM head operation in Eq. (2), causing significant overhead for each token generation as shown in Figure 3(a). However, since the exit window is selected as the interval where the prediction distribution already appears sufficiently decisive in offline calibration, only the top few candidates capture most of the information needed for exit decisions in that interval. Accordingly, we introduce a proxy LM head that reduces $W_{\text{lm}} \in \mathbb{R}^{V \times d}$ in Eq. (2) to top-$K$ candidates. Specifically, at the window starting layer $l_s$, we execute the full LM head operation once to extract the index set of the top $K$ tokens $\mathcal{I}_K = \text{top-}K(W_{\text{lm}} \mathbf{h}_{l_s})$, and construct $W_{\text{proxy}} = W_{\text{lm}}[\mathcal{I}_K, :] \in \mathbb{R}^{K \times d}$ by selecting only the corre-

sponding rows. For layers $l > l_s$, we compute logits and softmax margin using $W_{\text{proxy}}$ instead of $W_{\text{lm}}$ to determine whether to exit, and if the margin condition $\text{Margin}_l \geq \tau_m$ is satisfied, exit at that layer and output the top-1 token as the prediction result. In terms of computational overhead, the full LM head requires $2Vd$ FLOPs while the proxy LM head requires $2Kd$ FLOPs, achieving a reduction of $1 - K/V$. For example, using $K = 4096$ in LLaMA-2 7B can reduce head computation cost by approximately 87%. Additionally, within the exit window, the full LM head operation is performed only once at the starting layer $l_s$, and is replaced by the proxy LM head for up to $W - 1$ subsequent layers. This is based on the observation that since the window interval is already a region where high confidence is formed, the top candidate set obtained at $l_s$ is mostly maintained in subsequent layers. Therefore, the effectiveness of the proxy LM head is determined by whether the correct token is included in the top-$K$ candidate set.

As shown in the calibration results in Figure 5, using $K = 4096$ at layer 18 with 500 samples, the probability that the correct token is included in the top-$K$ candidate set was high at 99.6% for MMLU and 92.2% for CSQA, while BoolQ and SST2 recorded 100% coverage. This means that the top candidate tokens at the exit window starting point contain the correct answer in most cases, and the information needed for exit decisions can be sufficiently preserved even with a limited vocabulary. On the other hand, when the correct token is not included in the top-$K$ set, the proxy LM head's margin does not become sufficiently large to satisfy the exit condition, and the model computes through the final layer and performs prediction with the full LM head. Consequently, the proxy LM head reduces computation for confident predictions while computing through the original model path for uncertain cases, minimizing accuracy loss.

### 3.4. Training-free Deployment

Unlike SD, which requires additional training, WAVE adopts a fully training-free approach, constructing its exit policy using only lightweight calibration forward passes. Calibration proceeds in two stages. First, the model performs full forward passes over $M$ calibration prompts while recording the softmax margin at each layer. From these statistics, WAVE derives the hit layer distribution—defined as the earliest layer satisfying the margin threshold $\tau_m$—and identifies the interval where this distribution is concentrated as the exit window $\mathcal{W}^\star$. In parallel, the top-$K$ vocabulary at the window's starting layer $l_s$ is extracted to build the proxy LM head. Throughout this process, model parameters remain unchanged, and calibration is completed using only forward passes over a few thousand tokens. In contrast, SD-based methods require training draft models or auxiliary heads, incurring substantially higher offline costs due to repeated forward and backward passes.



*Figure 5.* Top-$K$ coverage of the correct answer token at layer $l$=18 with $K$=4096 on MMLU, CSQA, SST2 (Socher et al., 2013), and BoolQ (Clark et al., 2019). *In/Out* denotes whether the correct token is included in the top-$K$ set.

$$\frac{F_{\text{train}}}{F_{\text{cal}}} = \frac{6N_{\text{param}}^{D} \cdot N_{\text{tok}}}{2N_{\text{param}}^{T} \cdot T_{\text{cal}}} \qquad (5)$$

Draft model training requires approximately $6N_{\text{param}}$ FLOPs per token due to forward and backward passes, whereas WAVE's calibration incurs only $2N_{\text{param}}$ FLOPs, as it consists solely of forward passes. Compared to EAGLE (Li et al., 2024), which trains on 68K dialogues (100M tokens), WAVE reduces offline cost by over $10^4\times$. In practice, EAGLE's draft model requires approximately 24 hours of training on an RTX 3090 for LLaMA-2 7B (Xu et al., 2025), while WAVE completes calibration within minutes using only a few thousand tokens. This training-free characteristic provides three advantages in practical deployment environments. First, the exit policy can be rapidly reconfigured for new domains via lightweight calibration. Second, WAVE introduces no additional parameters or KV-cache overhead, as it requires no draft model or auxiliary heads. Third, it preserves a single-model structure, enabling seamless integration with existing serving pipelines and compatibility with standard optimizations such as quantization.

## 4. Experimental Results

### 4.1. Experimental Setup

To evaluate the effectiveness of WAVE, we conduct experiments on a diverse set of LLM benchmarks using LLaMA-2 7B Chat in a PyTorch environment on an NVIDIA RTX 4090 GPU. We evaluate knowledge and reasoning on MMLU (Hendrycks et al., 2020) and CSQA (Talmor et al., 2019), binary question answering on BoolQ (Clark et al., 2019), sentiment classification on SST2 (Socher et al., 2013), natural language inference on RTE (Wang et al., 2018), and semantic equivalence on MRPC (Wang et al., 2018).

All experiments are performed under FP16, with additional evaluation in a W4A16 quantized setting to verify quantization compatibility. We set the exit window size to $W = 8$,

margin threshold to $\tau_m = 0.2$, and proxy LM head size to $K = 4096$. As baselines, we include a dense decoding baseline (*i.e.*, LLaMA-2 7B Chat without early exit), along with AdaInfer (Fan et al., 2024) and SpecEE (Xu et al., 2025), which represent the most training-efficient among existing EE methods, despite WAVE being entirely training-free. AdaInfer performs per-layer exit decisions based on full LM head confidence, and SpecEE reduces the vocabulary search space via a trained draft model.

For the main results on LLaMA-2 7B, the exit window and proxy LM head are calibrated using samples from GSM8K, which is intentionally drawn from a different domain than the evaluation tasks to provide a practical assessment of WAVE's robustness to distribution mismatch.

### 4.2. Main Results

**Accuracy Preservation in FP16 model:** Table 1 summarizes the impact of WAVE on average layer usage and accuracy under the FP16 setting. On MMLU, WAVE improves accuracy by +0.12%p while using only 23.65 layers on average, reducing approximately 26% of the total 32 layers. On CSQA, it incurs a marginal accuracy drop of -0.09%p with a 27% layer reduction. For binary classification tasks, WAVE trades slight accuracy degradation for larger efficiency gains: on SST2 and BoolQ, it reduces layer usage by 36% and 38%, respectively, with accuracy drops below 0.7%p. On MRPC and RTE, WAVE achieves 70.59% and 76.53% accuracy, respectively, surpassing the prior state-of-the-art SpecEE by +1.82%p and +1.56%p, while still reducing computation by using 35% fewer layers on average (20.67/20.58 vs. 26.89/25.76). These gains, particularly on MMLU, MRPC, and RTE, are consistent with prior observations (Skean et al., 2025) that intermediate layers in autoregressive LLMs can yield more informative representations than the final layer, suggesting that early exit may offer benefits beyond computational savings. Compared to prior EE methods, AdaInfer preserves accuracy on MMLU but uses significantly more layers (28.91 on average), limiting efficiency, and suffers a larger drop on CSQA. SpecEE achieves slightly lower layer usage than WAVE but incurs greater accuracy degradation across tasks. Overall, WAVE strikes a superior balance between efficiency and accuracy, substantially reducing computation while maintaining—or even improving—task performance relative to existing EE approaches.

**Speedup in FP16 model:** While EE methods reduce computation by skipping layers, the resulting latency gains are not strictly proportional to the layer reduction due to the overhead of exit decisions. To evaluate practical efficiency, Table 2 reports the average end-to-end inference latency across datasets. The proposed method achieves speedups of 1.32× on MMLU, 1.31× on CSQA, and 1.57× on SST2,

*Table 1.* Evaluation of accuracy (higher is better) and the average number of layers used (#Avg. L, lower is better) on LLaMA-2 7B.

| Tasks | MMLU | | CSQA | | SST2 | | BoolQ | | MRPC | | RTE | |
|---|---|---|---|---|---|---|---|---|---|---|---|---|
| | Acc.↑ | #Avg. L↓ | Acc.↑ | #Avg. L↓ | Acc.↑ | #Avg. L↓ | Acc.↑ | #Avg. L↓ | Acc.↑ | #Avg. L↓ | Acc.↑ | #Avg. L↓ |
| Dense | 45.30 | 32.00 | 61.43 | 32.00 | 86.24 | 32.00 | 82.81 | 32.00 | 68.87 | 32.00 | 75.09 | 32.00 |
| AdaInfer | + 0.68 | 28.91 | - 0.50 | 27.90 | + 0.10 | 20.57 | - 0.37 | 19.52 | – | – | – | – |
| SpecEE | - 0.66 | 23.16 | - 0.17 | 22.90 | - 0.35 | 23.55 | 0.00 | 20.98 | - 0.10 | 26.89 | - 0.12 | 25.76 |
| Ours | + 0.12 | 23.65 | - 0.09 | 23.34 | - 0.45 | 20.34 | - 0.64 | 19.96 | + 1.72 | 20.67 | + 1.44 | 20.58 |

*Table 2.* End-to-end speedup on LLaMA-2 7B (FP16).

| Method | MMLU | CSQA | SST2 |
|---|---|---|---|
| Baseline | ×1.00 | ×1.00 | ×1.00 |
| AdaInfer | ×1.02 | ×1.01 | ×1.04 |
| SpecEE | ×1.30 | ×1.31 | ×1.28 |
| Ours | ×1.32 | ×1.31 | ×1.57 |

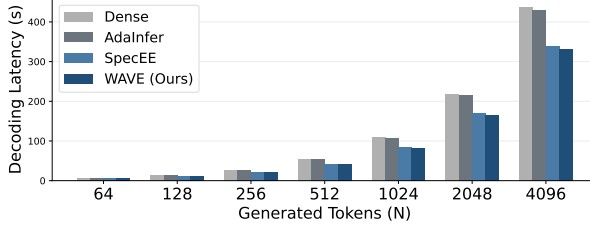

*Figure 6.* End-to-end decoding latency on LLaMA-2 7B across different generation lengths.

*Table 3.* Accuracy and speedup results on LLaMA-2 7B under W4A16 quantization.

| Method | Task | Acc.↑ | Avg.L↓ | Speedup↑ |
|---|---|---|---|---|
| Ours (BnB) | MMLU | 43.88 / 43.85 | 32.00 / 23.44 | 1.30× |
| | CSQA | 56.60 / 56.00 | 32.00 / 23.21 | 1.31× |
| | SST2 | 85.78 / 84.98 | 32.00 / 20.32 | 1.56× |
| Ours (AWQ) | MMLU | 43.43 / 43.12 | 32.00 / 25.07 | 1.29× |
| | CSQA | 56.66 / 56.87 | 32.00 / 24.73 | 1.27× |
| | SST2 | 85.55 / 84.98 | 32.00 / 20.77 | 1.49× |
| SpecEE (AWQ) | MMLU | 43.43 / 43.27 | 32.00 / 23.27 | 1.28× |
| | CSQA | 56.66 / 57.40 | 32.00 / 22.94 | 1.29× |
| | SST2 | 85.55 / 85.55 | 32.00 / 22.81 | 1.28× |

*Note:* Each cell shows Baseline / +EE.

yielding an overall average speedup of 1.4×. The largest improvement is observed on SST2, consistent with Table 1, where SST2 exhibits the lowest average layer usage (20.34). These results confirm that the proposed exit window scheduling and proxy LM head effectively translate reduced computation into tangible end-to-end latency improvements.

Figure 6 illustrates how inference overhead accumulates as generation length increases. Using a fixed 128-token prompt, we vary the number of generated tokens $N \in \{64, 128, 256, 512, 1024, 2048, 4096\}$ and measure end-to-end decoding latency. As $N$ grows, repeated exit evaluations introduce increasing overhead. AdaInfer, which evaluates logits at every token, achieves only 1.02× speedup for long sequences. SpecEE attains a 1.29× speedup via a predictor but incurs additional memory overhead from the draft model. In contrast, WAVE confines exit decisions to a calibrated window and employs a proxy LM head, achieving a 1.32× speedup with minimal overhead. At $N = 4096$, WAVE reduces latency to 331.8s, compared to 437.9s for dense decoding and 429.3s for AdaInfer (24.2% reduction). SpecEE records 339.2s, remaining 2.2% slower than WAVE due to draft model overhead. Overall, WAVE delivers faster inference without auxiliary models or training, reducing latency, memory usage, and deployment complexity, and thus offering a more practical solution for production settings.

**Compatibility with Quantization (W4A16):** Given that modern LLMs are typically deployed with quantized parameters, we evaluate the practicality of WAVE under widely used low-precision quantization schemes. Table 3 reports results under BitsAndBytes (BnB) (Dettmers et al., 2023) and Activation-Aware Weight Quantization (AWQ) (Lin et al., 2024). Under BnB, WAVE achieves 43.85% on MMLU (-0.03%p), 56.00% on CSQA (-0.60%p), and 84.98% on SST2 (-0.80%p), using on average 23.44, 23.21, and 20.32 layers, respectively. This closely matches FP16 behavior and yields end-to-end speedups of 1.30×−1.56×, demonstrating that the EE policy remains stable even under 4-bit weight quantization. Similar trends are observed under AWQ, where WAVE attains 43.12% on MMLU (-0.31%p), 56.87% on CSQA (+0.21%p), and 84.98% on SST2 (-0.57%), with average layer usage of 25.05, 24.73, and 20.77 and speedups of 1.27×−1.49×. Compared to SpecEE under AWQ, which uses fewer layers but achieves only 1.28×−1.29× speedup, WAVE delivers higher actual acceleration—most notably 1.49× on SST2—without requiring a draft model or additional training. Overall, WAVE's training-free, inference-only design ensures strong compatibility and deployability across diverse quantization settings.

**Generalization Across Model Architectures and Scales:** To validate the generalizability of WAVE, we conduct additional experiments on Mistral-7B (Jiang et al., 2023), Qwen2-7B (Yang et al., 2025), and LLaMA-2-13B. As shown in Table 4, WAVE consistently reduces the average number of executed layers across diverse model architec-

*Table 4.* Evaluation of accuracy (higher is better) and the average number of layers used (#Avg. L, lower is better) across diverse model architectures and scales.

| Model | MMLU | | CSQA | | SST2 | | BoolQ | | MRPC | | RTE | |
|---|---|---|---|---|---|---|---|---|---|---|---|---|
| | Acc.↑ | #Avg. L↓ | Acc.↑ | #Avg. L↓ | Acc.↑ | #Avg. L↓ | Acc.↑ | #Avg. L↓ | Acc.↑ | #Avg. L↓ | Acc.↑ | #Avg. L↓ |
| Mistral-7B | 59.32 | 32.00 | 71.74 | 32.00 | 93.92 | 32.00 | 86.02 | 32.00 | 77.45 | 32.00 | 80.51 | 32.00 |
| W/ Ours | 58.65 | 25.65 | 71.25 | 23.67 | 93.46 | 22.14 | 81.65 | 25.23 | 74.02 | 25.46 | 81.23 | 25.76 |
| Qwen2-7B | 74.25 | 28.00 | 84.68 | 28.00 | 94.38 | 28.00 | 86.88 | 28.00 | 74.51 | 28.00 | 81.59 | 28.00 |
| W/ Ours | 73.89 | 25.03 | 84.44 | 24.01 | 94.50 | 25.07 | 87.43 | 25.05 | 75.25 | 25.02 | 81.59 | 25.47 |
| LLaMA-2-13B | 53.58 | 40.00 | 67.90 | 40.00 | 94.15 | 40.00 | 84.65 | 40.00 | 73.04 | 40.00 | 74.37 | 40.00 |
| W/ Ours | 52.53 | 34.55 | 66.91 | 23.67 | 94.38 | 21.17 | 84.65 | 28.23 | 71.57 | 28.07 | 74.37 | 37.11 |

tures and scales while preserving accuracy at a level comparable to the dense baseline across most tasks. Specifically, WAVE reduces the average layer usage from 32.00 to 24.65 on Mistral-7B, from 28.00 to 24.94 on Qwen2-7B, and from 40.00 to 28.80 on LLaMA-2-13B, corresponding to average layer reductions of 22.96%, 10.92%, and 28.00%, respectively. Notably, we observe a general trend in which models with more transformer layers can provide greater room for layer reduction through exit window scheduling, although the magnitude of the gain varies across tasks and architectures. This tendency suggests that WAVE can remain effective under the prevailing scaling trend of LLMs, where both model capacity and depth continue to increase.

**Throughput under Batched Serving:**   To verify the effectiveness of WAVE in realistic serving environments, we additionally measure throughput under batched inference using vLLM (Kwon et al., 2023). As shown in Table 5, WAVE improves throughput over the dense baseline across all six tasks—MMLU, CSQA, SST2, BoolQ, MRPC, and RTE—with an average speedup of $1.13\times$.

In dynamic batching environments such as vLLM, the throughput gain is relatively modest compared to the single-request setting. This can be attributed to two factors: different exit layers across sequences in the same batch reduce the effective batch size at deeper layers, and vLLM is already highly optimized for batched inference. Meanwhile, in deployment scenarios where the batch size is close to 1, such as edge devices, single-request latency can be a more critical performance metric, and the latency reduction achieved by WAVE offers more pronounced practical benefits.

### 4.3. Overhead Analysis

**Theoretical FLOPs Breakdown:**   The efficiency of EE methods depends not only on reducing base computation but also on exit-decision overhead. AdaInfer invokes the full LM head at every layer, incurring $2Vd$ FLOPs per call—up to 8.4 GFLOPs/token on LLaMA-2 7B. As shown in Figure 7, with an average exit depth of $\bar{L} = 24$, this results in about 6.30 GFLOPs/token of overhead. SpecEE reduces this cost by limiting the vocabulary using an EAGLE-based predictor,

*Table 5.* Batched inference throughput (tokens/s) and average latency (ms/token) on LLaMA-2 7B using vLLM.

| Task | Throughput (tok/s)↑ | | Latency (ms/tok)↓ | |
|---|---|---|---|---|
| | Dense | Ours | Dense | Ours |
| MMLU | 101.94 | 107.66 | 9.82 | 9.29 |
| CSQA | 170.76 | 183.41 | 5.89 | 5.45 |
| SST2 | 244.49 | 305.55 | 4.09 | 3.27 |
| BoolQ | 59.47 | 62.22 | 16.82 | 16.07 |
| MRPC | 123.65 | 156.38 | 8.09 | 6.39 |
| RTE | 105.29 | 112.61 | 9.56 | 8.88 |

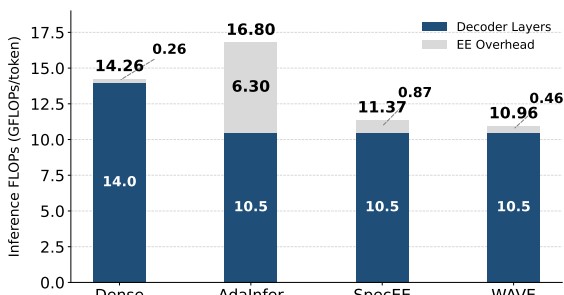

*Figure 7.* Comparison of per-token FLOPs on LLaMA-2 7B across Dense, AdaInfer, and WAVE.

but still incurs approximately 0.87 GFLOPs/token due to its auxiliary components. In contrast, WAVE performs the full LM head only once per 8-layer exit window and uses a lightweight proxy LM head thereafter, resulting in just 0.46 GFLOPs/token. Consequently, at $\bar{L} = 24$, WAVE reduces overhead by 5.84 GFLOPs/token compared to AdaInfer and by 0.41 GFLOPs/token compared to SpecEE, demonstrating that minimizing per-layer LM head evaluations is critical for realizing practical EE speedups.

**Computation Overhead Breakdown:**   Figure 8a reports exit-decision overhead on SST2 as a fraction of total inference time. WAVE completes inference in 21s with only 2.8% overhead. SpecEE requires 23.5s with 7.2% overhead, while AdaInfer incurs the highest overhead (34.1%) and latency (28.9s) due to repeated full LM head evaluations, despite

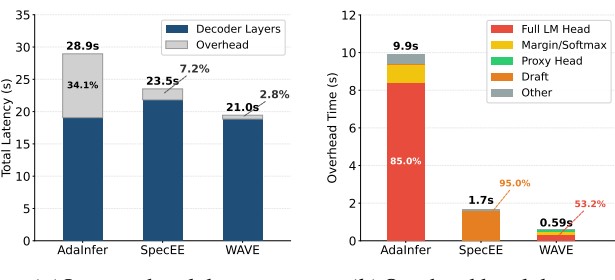

*(a)* Latency breakdown.     *(b)* Overhead breakdown.

*Figure 8.* End-to-end latency and overhead breakdown on LLaMA-2 7B, comparing WAVE and AdaInfer.

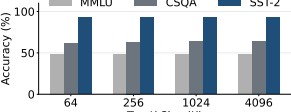
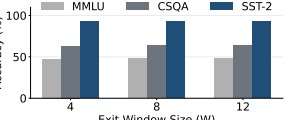

*(a)* Proxy LM head size $K$.     *(b)* Exit window size $W$.

*Figure 9.* Ablation study of WAVE on LLaMA-2 7B.

using a similar number of layers. Figure 8b further breaks down this overhead. AdaInfer's 9.9s overhead is dominated by full LM head calls (85%). SpecEE avoids full heads but relies on a draft model, which accounts for 95% of its 1.7s overhead. In contrast, WAVE incurs only 0.59s overhead, with just 16.4% from a single LM head evaluation, while the remainder is handled by lightweight proxy heads and margin scoring. By restricting exit evaluation to a narrow window and replacing costly operations with efficient approximations, WAVE reduces cumulative overhead by 94% relative to AdaInfer, ensuring that layer reductions translate into tangible inference speedups without auxiliary models.

### 4.4. Ablation Study

The proxy LM head vocabulary size $K$ directly affects the reliability of exit decisions. A larger $K$ increases the probability that the correct token is included in the top-$K$ candidate set, yielding confidence estimates more aligned with the full LM head. Conversely, a smaller $K$ may exclude the correct token, leading to unstable confidence estimation and more frequent fallback to full-depth computation. Figure 9a reports accuracy for $K \in \{64, 256, 1024, 4096\}$. CSQA shows improvement as $K$ increases, likely because its five answer choices require broader candidate coverage. This suggests that $K$ becomes more important as the number of answer candidates grows. Overall, $K = 4096$ provides a reliable default with 87% overhead reduction compared to the full LM head.

In addition, the exit window size $W$ defines the layer interval for exit evaluation, affecting both accuracy and overhead. A small $W$ reduces exit checks but may miss the confidence transition region identified during calibration, while a larger $W$ provides broader coverage at the cost of additional evaluations. Figure 9b shows accuracy for $W \in \{4, 8, 12\}$. Performance remains stable across all settings with minimal variation, indicating that WAVE is robust to the choice of $W$. However, since overhead scales linearly with $W$, selecting an appropriate value requires balancing coverage and computational cost. In our experiments, $W = 8$ offers an

effective trade-off, providing sufficient coverage for stable early exit while limiting evaluation overhead.

## 5. Conclusion

In this paper, we propose WAVE, a training-free early-exit framework that overcomes three key limitations of prior approaches: LM head overhead, generation length constraints, and training cost. WAVE introduces exit window scheduling, which identifies optimal exit intervals through lightweight offline calibration, eliminating per-layer exit checks while avoiding the premature convergence issues of schedule-based methods. In addition, a proxy LM head constructed from a top-$K$ vocabulary at the window's entry layer reduces computation by approximately 87% compared to full LM head evaluations. WAVE requires no retraining and readily adapts to new domains using only minimal calibration data. Experiments on LLaMA-2 7B demonstrate an average $1.4\times$ speedup on MMLU, CSQA, and SST2 with negligible accuracy degradation. Comparable gains are observed under W4A16 quantization (BitsAndBytes, AWQ), confirming compatibility with common LLM optimization techniques. Overall, WAVE provides a practical, drop-in solution for accelerating real-world LLM inference without architectural modification or additional training.

## Acknowledgements

This work was partly supported by Institute of Information & communications Technology Planning & Evaluation (IITP) grant funded by the Korea government(MSIT) (No.2020-0-01305, Development of AI Deep-Learning Processor and Module for 2,000 TFLOPS Server) and the Institute of Information & Communications Technology Planning & Evaluation(IITP) grant funded by the Korea government(MSIT) (No. RS-2026-25519380).

## Impact Statement

This paper presents work whose goal is to advance the field of Machine Learning. There are many potential societal consequences of our work, none which we feel must be specifically highlighted here.

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

## A. Long-form Generation Results

To complement the short-form evaluation in the main experiments, we conduct additional experiments on GSM8K and TruthfulQA_gen (Lin et al., 2022). As shown in Table 6, WAVE achieves approximately 1.1× speedup on GSM8K and 1.14× on TruthfulQA_gen.

To further examine behavior under ultra-long context settings, we analyze the first-hit layer distribution by calibrating on 100 samples from LongBench-v2 (Bai et al., 2025). The average first-hit layer is 24.9, with most first hits concentrated in layers 23–26, suggesting that the calibrated exit window remains structurally meaningful even under long-context inputs.

However, the accuracy degradation observed on GSM8K suggests that while the exit window placement remains valid, the KV cache inconsistency introduced by early exit poses a remaining challenge for long-form generation. We hypothesize that this is attributable to a structural characteristic of decoder-only architectures. In encoder-decoder models, prefill and decoding are handled by separate blocks, so early exit during decoding does not affect the encoder representations of the input prompt. In contrast, decoder-only models share a single layer stack for both the input prompt and generated tokens, with prefill information retained through the KV cache. Consequently, when early exit prevents certain layers from updating their KV cache entries, accumulated representation loss occurs over long generation sequences. We expect this limitation to be addressed in future work through exit policies that explicitly account for KV cache consistency.

*Table 6.* Accuracy and average number of layers used on long-form generation tasks.

| Task | Acc.↑ | | #Avg. L↓ | |
|---|---|---|---|---|
| | Dense | +WAVE | Dense | +WAVE |
| GSM8K | 21.76 | 19.99 | 32.00 | 27.83 |
| TruthfulQA_gen | 44.92 | 45.17 | 32.00 | 27.93 |

## B. Sensitivity Analysis on $\tau$

To analyze the sensitivity of the exit threshold $\tau$, we conduct additional experiments over $\tau \in \{0.3, 0.5, 0.7\}$. As shown in Table 7, accuracy remains stable across all settings at 63.88 / 63.55 / 63.72, despite varying degrees of exit aggressiveness. A smaller $\tau$ leads to more aggressive early exit and lower average layer usage, while a larger $\tau$ results in more conservative exit behavior. The overall stability in accuracy can be attributed to the fact that tokens for which the model is not confident tend to receive probabilities close to zero, meaning $\tau$ acts as a gentle acceptance criterion for confident tokens rather than a fragile threshold requiring precise tuning. Furthermore, directly applying $\tau$ calibrated under FP16 to a W4A16 quantized setting without re-tuning yields stable performance, demonstrating the quantization compatibility of the threshold.

*Table 7.* Sensitivity analysis on exit threshold $\tau$ on CSQA with LLaMA-2 7B.

| $\tau$ | Acc.↑ | #Avg. L↓ |
|---|---|---|
| 0.3 | 63.88 | 25.91 |
| 0.5 | 63.55 | 27.00 |
| 0.7 | 63.72 | 28.07 |

