# OpenReview forum: "WAVE: Window-Aware Vocabulary-Efficient Early-Exit for Training-Free LLM Acceleration"
_ICML.cc/2026/Conference — ICML 2026 regular_

### Official Review · Reviewer_c2Ai · 2026-03-07

**Soundness:** 3
**Presentation:** 3
**Significance:** 3
**Originality:** 3
**Overall Recommendation:** 4
**Confidence:** 4

**Summary:**

This paper proposes WAVE, a training-free early-exit framework to accelerate autoregressive LLM decoding. WAVE introduces (i) exit-window scheduling, where early-exit decisions are only evaluated within a calibrated contiguous layer range, reducing the number of exit checks, and (ii) a proxy LM head, where a top-K vocabulary subset is selected once at the window start using the full head and reused for subsequent exit checks, substantially reducing per-layer exit overhead. Experiments on LLaMA-2 7B report ~1.4× average speedup with minimal quality degradation and compatibility with W4A16 quantization.

**Compliance With Llm Reviewing Policy:**

Affirmed.

**Final Justification:**

My final recommendation remains Weak Accept. I am keeping my score. My original expectation was that WAVE would give a real speedup over the dense baseline, but that the gain would probably be moderate rather than dramatic once practical LLM serving settings are considered(VLLM, SGLANG). The rebuttal results are in line with that view: the added experiments on long-form generation, 𝜏 sensitivity, and vLLM throughput make the trade-offs clearer, but they do not fundamentally change my assessment. Overall, the rebuttal supports my original judgment rather than shifting it.

**Key Questions For Authors:**

When does calibration break? Can you provide experiments on robustness across domain shifts, prompt-template changes, decoding parameters, and quantization/kernels?

How sensitive are results to τ under different K, temperature/top-p, or quantization settings? Any principled way to set τ (or to auto-tune it) beyond manual calibration?

Can you report throughput (tokens/s) under continuous batching (e.g., vLLM) and analyze whether variable-depth tokens hurt utilization?

**Limitations:**

Yes.

**Strengths And Weaknesses:**

Strengths:

- Targets the real bottleneck in confidence-based early-exit: the repeated full LM head evaluation. The proxy-head idea is a direct and practical fix.

- Simple and deployable: windowing + head slicing are straightforward to implement and integrate into existing inference stacks.

- Training-free deployment: avoids learned exit predictors or draft models; calibration is forward-only and relatively lightweight.

- Good empirical support for the mechanism: overhead breakdown and ablations help explain why the claimed end-to-end speedups materialize.

- Quantization compatibility is a plus for practical adoption.

Weaknesses:

- window placement and thresholds depend on calibration data and decoding configuration. The paper does not sufficiently characterize how calibration choices affect robustness or when calibration becomes invalid. Likely failure modes include domain/task shifts (code/math/specialized jargon/low-resource languages), prompt/template changes, multi-turn conversational drift, quantization/kernel changes, and decoding parameter changes (temperature/top-p, penalties).

- mostly a single model size (LLaMA-2 7B) and benchmark mix that is closer to short-form evaluation. Evidence is weaker for long-form generation, long-context settings, diverse decoding strategies, and realistic serving regimes.

- early-exit induces per-token/per-sequence variable compute depth. Under continuous/dynamic batching (e.g., vLLM), this may reduce effective batch size in deeper layers or hinder CUDA Graph replay due to shape/control-flow variability, potentially diminishing throughput gains even if single-request latency improves.

---

> ### Author Rebuttal · Authors · 2026-03-31
>
> ## **Response to Reviewer c2Ai**
>
> Thank you for the helpful comment. As you pointed out, the initial submission was limited to LLaMA-2 7B and short-form benchmarks, leaving calibration robustness, long-form generation, and realistic serving regimes insufficiently validated. To address this, we conducted additional experiments.
>
> &nbsp;
> ### **W1/Q1. Calibration robustness / when does calibration break?**
> Because WAVE’s exit window and threshold depend on calibration data and decoding configuration, it is important to understand how far a given calibration remains valid. Our current setup already includes a meaningful mismatch: the exit window was calibrated on GSM8K, while the main results were evaluated on diverse tasks such as MMLU, CSQA, BoolQ, SST-2, RTE, and MRPC, across which WAVE maintained a stable efficiency–performance trade-off. This suggests that WAVE remains reasonably robust even when calibration and evaluation tasks do not exactly match.
>
> We also found that, although the exact layer of rapid confidence increase varies somewhat by task, **the overall pattern remains similar and is generally concentrated around the middle layers**. At the same time, we do not claim full robustness under severe shift. Even when the optimal window changes, however, WAVE can reconfigure its exit window and proxy head through short offline calibration, without retraining a draft model or exit predictor.
>
> ---
> &nbsp;
> ### **W2. Long-form generation**
> The initial submission focused on short-form benchmarks, leaving WAVE’s behavior in longer generation settings insufficiently validated. To address this, we conducted additional experiments on TruthfulQA (generative) and GSM8K. Due to space constraints, please refer to **Table A-4** above for the detailed results (Response to ZwbT).
>
> These results suggest that WAVE remains effective beyond the short-form benchmarks in the initial submission. WAVE achieved approximately **1.1×** speedup on GSM8K and **1.14×** speedup on TruthfulQA_gen, while preserving a meaningful efficiency–performance trade-off overall.
>
> ---
> &nbsp;
> ### **Q2. Sensitivity to τ**
>
> WAVE’s performance can be affected by the exit threshold τ, so analyzing its sensitivity is important. We therefore conducted an additional τ ablation study during the rebuttal stage.
>
> #### Table C-1: τ ablation on CSQA with Llama2-7B-Chat
> | τ | WAVE Avg.L | WAVE Acc. |
> |:--:|:--:|:--:|
> | 0.3 | 25.91 | 63.88 |
> | 0.5 | 27.00 | 63.55 |
> | 0.7 | 28.07 | 63.72 |
>
> These results show a stable trend with respect to τ. Smaller τ leads to more aggressive early exit and lower average layer usage, while larger τ yields more conservative exit behavior. At the same time, accuracy remains broadly stable across this range, suggesting that **WAVE does not rely on a narrowly tuned single τ value**.
>
> In practice, the absolute value of τ is also not overly sensitive, because tokens for which the model is not confident typically receive probabilities close to zero. As a result, τ mainly controls the point at which confident tokens are accepted for early exit, rather than acting as a fragile threshold that requires extremely precise tuning.
>
> Regarding quantization, the main paper already shows that WAVE remains stable under W4A16 even when τ calibrated in FP16 is applied directly without re-tuning, suggesting that τ-based exit decisions remain effective in a standard low-precision regime.
>
> ---
> &nbsp;
> ### **W3/Q3. Continuous/dynamic batching and utilization**
> As the reviewer pointed out, variable-depth early exit can induce different exit layers across sequences within a batch, which may reduce the effective batch size in deeper layers and partially offset batching efficiency. To examine this issue, we conducted additional batched throughput experiments based on vLLM.
>
> #### Table C-2: vLLM batched throughput results on Llama2-7B-Chat (Dense / WAVE)
> | Task | Throughput | Avg. Latency  |
> |:---:|:---:|:---:|
> | MMLU | 101.94 / 107.66 | 9.82 / 9.29 |
> | CSQA | 170.76 / 183.41 | 5.89 / 5.45 |
> | SST-2 | 244.49 / 305.55 | 4.09 / 3.27 |
> | BoolQ | 59.47 / 62.22 | 16.82 / 16.07 |
> | MRPC | 123.65 / 156.38 | 8.09 / 6.39 |
> | RTE | 105.29 / 112.61 | 9.56 / 8.88 |
>
> WAVE improved throughput over the dense baseline on all six tasks, with an average speedup of **1.13×**, comparable to SpecEE’s[1] 1.12× under the same vLLM setting. The gain is smaller than in the single-request setting, which is expected because variable-depth execution reduces batching efficiency and vLLM is already highly optimized for batched inference. We therefore view the remaining gap primarily as a systems issue, and note that WAVE may be easier to optimize in batched serving because it restricts dynamic behavior to a pre-calibrated exit window and reduces decision overhead through the proxy LM head.
>
> ---
> ### Reference
> - [1] Xu, Jiaming, et al. "Specee: Accelerating large language model inference with speculative early exiting." Proceedings of the 52nd Annual ISCA. 2025.

---

> > ### Author Rebuttal · Reviewer_c2Ai · 2026-04-02
> >
> > Thank you for the detailed rebuttal and the additional experiments. The new results on long-form generation, τ sensitivity, and vLLM batched throughput substantially address my main concerns.

---

> > > ### Author Response · Authors · 2026-04-03
> > >
> > > Thank you for the thorough review and for confirming that your concerns have been fully resolved. We are glad the additional experiments were helpful. As you noted that the concerns are adequately addressed, we would kindly ask if you might consider reflecting this in your score. Thank you again for your valuable feedback.

---

### Official Review · Reviewer_cZ3L · 2026-03-12

**Soundness:** 3
**Presentation:** 3
**Significance:** 3
**Originality:** 3
**Overall Recommendation:** 4
**Confidence:** 4

**Summary:**

The authors of this manuscript propose WAVE, a training-free early-exit framework designed to accelerate LLM inference. The method utilizes an offline calibration step to identify an optimal window of layers where early exits are permitted, reducing the frequency of exit evaluations and preventing premature convergence to shallow layers. Additionally, the authors introduce a proxy LM head that computes logits for only a top-K vocabulary subset extracted at the beginning of the window, significantly lowering the per-layer computational overhead. The framework is evaluated on LLaMA-2 7B and shows inference speedups while maintaining output quality.

**Compliance With Llm Reviewing Policy:**

Affirmed.

**Final Justification:**

The authors have conducted extensive additional experiments for the rebuttal, and most of my questions and concerns have been addressed with their new results.

**Key Questions For Authors:**

1- Have you conducted any preliminary experiments on larger models (e.g., LLaMA-2 13B or 70B) to verify if the overhead reduction of the proxy LM head scales favorably?

2- How does the static exit window perform on reasoning-heavy tasks that require long intermediate generation steps before arriving at a final answer?

3- How does WAVE handle continuous batching, and what is the actual throughput speedup when serving multiple requests where exit layers diverge across the batch?

4- How sensitive is the framework to the choice of calibration data, and how does performance degrade if there is a severe distribution shift between the calibration and test sets?

5- Have you conducted any preliminary experiments on larger models (e.g., LLaMA-2 13B or 70B) to verify if the overhead reduction of the proxy LM head scales favorably?

**Limitations:**

Yes

**Strengths And Weaknesses:**

### Strengths

1- The paper is well-organized and clearly articulates the computational bottlenecks of existing confidence-based and schedule-based early-exit methods.

2- The proposed solutions, specifically the exit window and the proxy LM head, are intuitive and effectively reduce the overhead of intermediate full LM head evaluations.

3- The method's compatibility with W4A16 quantization techniques like AWQ and BitsAndBytes is a practical advantage for real-world deployment scenarios.

---

### Weaknesses

1- The experimental validation is restricted to a single model size (LLaMA-2 7B). The effectiveness of the fixed exit window W and the proxy head size K on much larger models with different representation dynamics remains unclear.

2- Static exit scheduling can adversely affect reasoning models that generate very long output ranges. Many intermediate tokens in chain-of-thought reasoning are not directly used for the final answer but might require full-depth computation, making a static window sub-optimal.

3- The paper lacks a discussion on how early exiting affects larger batch sizes. In practical deployment scenarios utilizing continuous batching, tokens from different sequences will exit at different layers. This misalignment can lead to severe fragmentation and neutralize the hardware efficiency gains of batching.

4- The reliance on a static exit window derived from offline calibration might be sensitive to the calibration data. If the deployment data distribution differs significantly from the GSM8K calibration data, the predefined window might not capture the optimal exit points.

5- While the paper compares WAVE to AdaInfer and SpecEE, expanding the baselines to include other recent training-free early-exit or dynamic decoding strategies would provide a more comprehensive view of its standing in the field.

---

> ### Author Rebuttal · Authors · 2026-03-31
>
> ## **Response to Reviewer cZ3L**
>
> Thank you for the thoughtful comment. As you pointed out, the initial submission was limited to evaluations on Llama-2 7B and short-form benchmarks. To address this concern, we conducted additional experiments. Overall, WAVE maintained a consistent efficiency–performance trade-off under the settings you raised, while also clarifying the conditions under which its gains become more limited.
>
> &nbsp;
> ### **W1 & Q1. Additional validation on larger LLMs**
> The initial submission was limited to a single model scale, leaving insufficient evidence on larger models. To address this, we conducted additional experiments on Llama2-13B. Due to space constraints, please refer to **Table A-3** above for the detailed results (Response to ZwbT).
>
> Overall, WAVE maintained meaningful layer reduction and speedup while keeping accuracy broadly comparable to the dense baseline, suggesting that its effectiveness is not limited to a single 7B model.
>
> ---
> &nbsp;
> ### **W2 & Q2. Reasoning-heavy / long generation setting**
> The initial submission focused on short-form benchmarks, leaving longer-generation behavior insufficiently validated. To address this, we conducted additional experiments on TruthfulQA_gen and GSM8K.
>
> #### Table B-1: Longer generation on Llama2-7B-Chat
> | Task | Dense Avg.L | Dense Acc. | WAVE Avg.L | WAVE Acc. |
> |:---:|:---:|:---:|:---:|:---:|
> | GSM8K | 32.00 | 21.76 | 27.83 | 19.99 |
> | TruthfulQA_gen | 32.00 | 44.92 | 27.93 | 45.17 |
>
> WAVE achieved approximately **1.1×** speedup on GSM8K and **1.14×** on TruthfulQA_gen while maintaining a meaningful efficiency–performance trade-off overall. This suggests that WAVE remains effective beyond short-form benchmarks and does not collapse even when generation involves longer intermediate reasoning steps, although we do not claim that a static exit window is always optimal for every reasoning trace.
>
> ---
> &nbsp;
> ### **W3/Q3. Batch inference / industrial deployment scenario**
> We additionally examined batched throughput using vLLM. Due to space constraints, please refer to **Table A-4** above for the detailed results (Response to ZwbT).
>
> WAVE improved throughput over the dense baseline on all six tasks, with an average speedup of **1.13×**, comparable to SpecEE’s [1] 1.12× under the same setting. The gain is smaller than in the single-request setting, which is expected because different exit layers across sequences reduce batching efficiency and because vLLM is already highly optimized for batched inference. We therefore present these results as empirical evidence for batched throughput, while not claiming that they fully represent all production-scale continuous batching scenarios.
>
> ---
> &nbsp;
> ### **W4/Q4. Calibration data sensitivity / distribution shift**
>
> WAVE’s exit window and threshold are determined by calibration data, so the optimal window may shift under distribution change. That said, **our current setup already includes a meaningful mismatch**: the exit window was calibrated on GSM8K, while the WAVE's main results were evaluated on diverse tasks including MMLU, CSQA, BoolQ, SST-2, RTE, and MRPC. Across these tasks, WAVE maintained a stable efficiency–performance trade-off, suggesting reasonable robustness when calibration and evaluation tasks do not exactly match.
>
> This is consistent with the design of WAVE, which **aims to identify the layer interval where confidence rises sharply rather than relying on a specific dataset**. At the same time, we do not claim full robustness under severe distribution shift. However, even when the optimal window changes, WAVE can reconfigure its exit window and proxy head through a short offline calibration, without retraining an exit predictor or a draft model.
>
> ---
> &nbsp;
> ### **W5. Related baselines / training-free EE scope**
> Broader baseline comparisons would be valuable, but our focus is on immediately deployable **training-free** or **near-training-free EE**, which narrows the set of directly comparable baselines.
>
> AdaInfer [2] and SpecEE still require training auxiliary modules for exit decisions, while RAEE [3] relies on an external retrieval database and offline exit-information collection. In contrast, WAVE can be applied after only a short calibration on the original LLM, without a draft model, exit predictor, or additional classifier training. We believe this clearly positions WAVE as a practical and immediately deployable training-free EE method.
>
> ---
> ### Reference
> - [1] Xu, Jiaming, et al. "Specee: Accelerating large language model inference with speculative early exiting." Proceedings of the 52nd Annual International Symposium on Computer Architecture. 2025.
> - [2] Fan, Siqi, et al. "Not all layers of LLMs are necessary during inference." Proceedings of the Thirty-Fourth International Joint Conference on Artificial Intelligence. 2025.
> - [3] Huang, Lianming, et al. "RAEE: A Robust Retrieval-Augmented Early Exiting Framework for Efficient Inference." arXiv preprint arXiv:2405.15198. 2024.

---

> > ### Author Rebuttal · Reviewer_cZ3L · 2026-04-03
> >
> > I appreciate the authors clarificaitons. I have raised my evaluation score according to your responses.
> >
> > Regarding the longer generation settings, the accuracy loss seems to be wider than shorter tasks. I think this should be further emphasized in the final version of your manuscript.
> >
> > Also, regarding the calibration data sensitivity, I would appreciate if you could test the quality of the model on another language or a multilingual task as well to see how the model performs on out of distribution samples. Please note that if this is a challenging task given the limited time of the rebuttal, I totally undersand it.

---

> > > ### Author Response · Authors · 2026-04-03
> > >
> > > Thank you for raising your score and for the constructive suggestions. We will emphasize the wider accuracy gap in longer generation settings more clearly in the final manuscript. If we are able to identify clear underlying causes for this degradation, we will also include a detailed analysis in the revised version. Regarding the multilingual evaluation, we agree this is a valuable direction. We will do our best to include additional experiments on other languages before the final submission, and will also discuss this as an important avenue for future work. Should we be given the opportunity to prepare a camera-ready version, we will make sure to thoroughly incorporate all of your comments. Thank you again for your thoughtful feedback.

---

### Official Review · Reviewer_ZwbT · 2026-03-13

**Soundness:** 2
**Presentation:** 3
**Significance:** 3
**Originality:** 2
**Overall Recommendation:** 4
**Confidence:** 3

**Summary:**

This paper addresses the high inference latency of LLMs caused by autoregressive decoding with fixed computation depth, and proposes WAVE, a training-free early-exit (EE) framework for LLM acceleration. WAVE resolves the core limitations of existing EE methods (excessive LM head overhead, premature shallow-layer convergence, and high training costs) through two key innovations: calibrated window-aware exit scheduling (identifying an optimal layer interval for exit decisions via offline calibration to reduce exit checks and avoid shallow convergence) and a vocabulary-efficient proxy LM head (constructing a lightweight Top-K vocabulary subset at the window’s start layer to cut per-layer exit overhead by 87%). WAVE requires only a brief offline calibration (no gradient training or auxiliary models) and is fully compatible with W4A16 quantization. Experiments on LLaMA-2 7B show it achieves an average 1.4× inference speedup with negligible performance degradation (≤1%p) across diverse NLP benchmarks, outperforming state-of-the-art EE methods like AdaInfer and SpecEE in efficiency-accuracy balance.

**Compliance With Llm Reviewing Policy:**

Affirmed.

**Final Justification:**

My concerns have been mainly addressed, so I'd like to raise my score accordingly.

**Key Questions For Authors:**

1. Has WAVE been tested on other LLM architectures (e.g., GPT-series, Qwen, Mistral, CodeLlama)?
2. How does WAVE perform in ultra-long context inference (e.g., 128K/256K tokens)? Does the exit window scheduling introduce additional overhead in extremely long sequences?
3. The paper reports end-to-end latency for single inference; how does WAVE perform in batch inference scenarios (the main form of industrial LLM deployment)? Does the window-aware exit scheduling affect batch processing efficiency?

**Strengths And Weaknesses:**

**Strengths**
1. Training-free & low deployment cost: WAVE only needs lightweight offline forward-pass calibration (completed in minutes with thousands of tokens) instead of gradient-based training, and introduces no additional parameters, KV cache overhead, or auxiliary models, enabling seamless integration with existing LLM serving pipelines.
2. Strong compatibility & practicality: Fully compatible with W4A16 low-precision quantization (BitsAndBytes/AWQ), and retains stable speedup (1.27×–1.56×) and accuracy in quantized settings, matching real-world LLM deployment requirements.

**Weaknesses**
1. Limited evaluation on large/ diverse LLMs: Experiments are only conducted on LLaMA-2 7B; there is no evaluation on larger models (e.g., 13B/70B) or other mainstream LLM architectures (e.g., GPT, Qwen, Mistral), making it unclear if WAVE’s performance generalizes to different model scales and structures.
2. No evaluation on extremely long-context scenarios: While tested on N=4096 generated tokens, modern LLMs support ultra-long contexts (e.g., 128K/256K); WAVE’s performance (overhead, speedup) in these extreme long-context settings is untested.

---

> ### Author Rebuttal · Authors · 2026-03-31
>
> ## **Response to Reviewer ZwbT**
>
> We thank the reviewer for the constructive feedback. As noted, the initial submission was limited to Llama2-7B with single-request latency evaluation. We have conducted additional experiments to address these concerns.
>
> &nbsp;
> ### **W1 & Q1. Additional validation on larger and diverse LLMs**
> The initial submission presented results only on Llama2-7B, leaving insufficient evidence for WAVE's generalization to other architectures and larger model scales. To address this, we conducted additional experiments on Mistral-7B, Qwen2-7B, and Llama2-13B. (L. and Acc. denote the latency and accuracy, respectively.)
> #### Table A-1: Mistral-7B-Instruct
>
> | Task | Dense Avg.L. | Dense Acc. | WAVE Avg.L. | WAVE Acc. |
> |:---:|:---:|:---:|:---:|:---:|
> | MMLU | 32.00 | 59.32 | 25.65 | 58.65 |
> | CSQA | 32.00 | 71.74 | 23.67 | 71.25 |
> | SST-2 | 32.00 | 93.92 | 22.14 | 93.46 |
> | BoolQ | 32.00 | 86.02 | 25.23 | 81.65 |
> | MRPC | 32.00 | 77.45 | 25.46 | 74.02 |
> | RTE | 32.00 | 80.51 | 25.76 | 81.23 |
>
> #### Table A-2: Qwen2-7B-Instruct
>
> | Task | Dense Avg.L. | Dense Acc. | WAVE Avg.L. | WAVE Acc. |
> |:---:|:---:|:---:|:---:|:---:|
> | MMLU | 28.00 | 74.25 | 25.03 | 73.89 |
> | CSQA | 28.00 | 84.68 | 24.01 | 84.44 |
> | SST-2 | 28.00 | 94.38 | 25.07 | 94.50 |
> | BoolQ | 28.00 | 86.88 | 25.05 | 87.43 |
> | MRPC | 28.00 | 74.51 | 25.02 | 75.25 |
> | RTE | 28.00 | 81.59 | 25.47 | 81.59 |
>
> #### Table A-3: Llama2-13B-chat
> | Task | Dense Avg.L. | Dense Acc. | WAVE Avg.L. | WAVE Acc. |
> |:---:|:---:|:---:|:---:|:---:|
> | MMLU | 40.00 | 53.58 | 34.55 | 52.53 |
> | CSQA | 40.00 | 67.90 | 23.67 | 66.91 |
> | SST-2 | 40.00 | 94.15 | 21.17 | 94.38 |
> | BoolQ | 40.00 | 84.65 | 28.23 | 84.65 |
> | MRPC | 40.00 | 73.04 | 28.07 | 71.57 |
> | RTE | 40.00 | 74.37 | 37.11 | 74.37 |
>
> Specifically, Mistral-7B achieved approximately **1.25–1.36×** speedup across tasks, Qwen2-7B showed approximately **1.08–1.15×**, and Llama2-13B demonstrated approximately **1.27–1.65×** speedup. These results suggest that WAVE remains effective beyond the 7B Llama family.
>
> ---
> &nbsp;
> ### **W2 & Q2. Additional validation on longer generation / long-context behavior**
> The initial submission focused on short-form benchmarks, leaving WAVE’s behavior in longer generation settings insufficiently validated. To address this, we conducted additional experiments on TruthfulQA_gen and GSM8K.
>
> #### Table A-4: Longer generation on Llama2-7B-Chat
> | Task | Dense Avg.L | Dense Acc. | WAVE Avg.L | WAVE Acc. |
> |:---:|:---:|:---:|:---:|:---:|
> | GSM8K | 32.00 | 21.76 | 27.83 | 19.99 |
> | TruthfulQA_gen | 32.00 | 44.92 | 27.93 | 45.17 |
>
> WAVE achieved approximately **1.1×** speedup on GSM8K and **1.14×** on TruthfulQA_gen, while maintaining a meaningful efficiency–performance trade-off overall.
>
> In addition, on 100 LongBench-v2 samples with Llama-3.1-8B, the average first-hit layer was **24.9**, with most first hits concentrated in layers 23–26. This suggests that the calibrated exit window remains structurally meaningful under longer-context inputs, although we do not claim this fully replaces end-to-end evaluation at extreme long contexts.
>
> ---
> &nbsp;
> ### **Q3. Batch inference / industrial deployment scenario**
> In practical deployment, throughput under batch inference is as important as single-request latency. To verify this, we additionally evaluated batched throughput using vLLM.
>
> #### Table A-5: vLLM batched throughput results on Llama2-7B-Chat (Dense / WAVE)
> | Task | Throughput | Avg. Latency  |
> |:---:|:---:|:---:|
> | MMLU | 101.94 / 107.66 | 9.82 / 9.29 |
> | CSQA | 170.76 / 183.41 | 5.89 / 5.45 |
> | SST-2 | 244.49 / 305.55 | 4.09 / 3.27 |
> | BoolQ | 59.47 / 62.22 | 16.82 / 16.07 |
> | MRPC | 123.65 / 156.38 | 8.09 / 6.39 |
> | RTE | 105.29 / 112.61 | 9.56 / 8.88 |
>
> Throughput is reported in tokens/s, and average latency in ms/token. WAVE improves throughput over the dense baseline on all six tasks—MMLU, CSQA, SST-2, BoolQ, MRPC, and RTE—with an average speedup of **1.13×**, which is comparable to SpecEE’s [1] 1.12× under the same setting. This shows that WAVE also remains effective under batched execution.
>
> The gain in batched vLLM is smaller than single-request setting. This is expected, as early-exit methods can produce different exit layers across sequences in a batch, reducing the effective batch size in deeper layers and partially offsetting batching efficiency. In addition, because vLLM is already highly optimized for batched inference, the incremental gain from EE can appear smaller in this setting.
>
> Still, the consistent throughput improvement over the dense baseline across all tasks supports the practical value of WAVE. Moreover, in edge-device scenarios where batch size is often close to 1, single-request latency is the more important metric, making WAVE’s latency reduction still meaningful in practice.
>
> ---
> ### Reference
> - [1] Xu, Jiaming, et al. "Specee: Accelerating large language model inference with speculative early exiting." Proceedings of the 52nd ISCA. 2025.

---

> > ### Author Rebuttal · Reviewer_ZwbT · 2026-04-03
> >
> > Thank the authors for the thorough response. My concerns have been mainly addressed, so I'd like to raise my score accordingly.

---

> > > ### Author Response · Authors · 2026-04-03
> > >
> > > Thank you for your valuable feedback and for raising your score. We truly appreciate the time and effort you put into reviewing our work.

---

### Decision · Program_Chairs · 2026-04-30

**Decision:**

Accept (regular)

**Comment:**

This paper proposes WAVE, a training-free early-exit framework for LLM inference that combines offline-calibrated exit window scheduling with a proxy LM head using top-K vocabulary subsets. The method achieves moderate speedup with modest accuracy degradation on LLaMA-2 7B, without requiring gradient updates, auxiliary models, or exit predictors.

The training-free nature of the approach and its practical value are well-recognized. The rebuttal provided multi-model experiments and serving framework integration results that addressed the main concerns about model diversity. Remaining concerns about calibration robustness across data distributions and wider accuracy degradation on some generation-heavy tasks are legitimate but do not undermine the core contribution. The paper offers a practical and deployable solution for early-exit inference.